META-RESEARCH ARTICLE

# Benchmarking with synthetic communities provides a baseline for virus-host inferences from Hi-C proximity linking

Rokaiya Nurani Shatadru [1,2], Natalie E. Solonenko[1,2], Christine L. Sun[2], Matthew B. Sullivan [1,2,3,4]*

1 Department of Microbiology, The Ohio State University, Columbus, Ohio, United States of America,
2 Center of Microbiome Science, The Ohio State University, Columbus, Ohio, United States of America,
3 Byrd Polar and Climate Research Center, The Ohio State University, Columbus, Ohio, United States of America, 4 Department of Civil, Environmental and Geodetic Engineering, The Ohio State University, Columbus, Ohio, United States of America

* sullivan.948@osu.edu

## Abstract

Microbiomes influence diverse ecosystems, and viruses increasingly appear to impose key constraints. While viromics has expanded genomic catalogs, host identification for these viruses remains challenging due to the limitations in scaling cultivation-based approaches and the uncertain reliability and relative low resolution of *in silico* predictions – particularly for understudied viral taxa. Towards this, Hi-C proximity ligation uses sequenced, cross-linked virus and host genomic fragments to infer virus-host linkages and has now been applied in at least 10 studies. However, its accuracy remains unknown. Here we assess Hi-C performance in recovering virus-host interactions using synthetic communities (SynComs) composed of four marine bacterial strains and nine phages with known interactions and then apply optimized bioinformatic protocols to natural soil samples. In SynComs, standard Hi-C sample preparations and analyses showed poor normalized contact score performance (26% specificity, 100% sensitivity, incorrect matches up to class level) that could be dramatically improved by $Z$-score filtering ($Z \geq 0.5$, 99% specificity), though at reduced sensitivity (62% down from 100%). Detection limits were established as reproducibility was poor below minimal phage abundances of $10^5$ PFU/mL. Applying optimized bioinformatic protocols to natural soil samples, we compared virus-host linkages inferred from proximity-ligated Hi-C sequencing with predictions generated by *in silico* homology-based and machine learning-based bioinformatic approaches. Prior to $Z$-score thresholding, agreement was relatively high at the phylum to family levels (72%), but not at the genus (43%) or species (15%) levels. $Z$-score thresholding reduced sensitivity (only 34% of predictions were retained), with only modest improvements in congruence with bioinformatic methods (48% or 18% at genus or species levels, respectively). Regardless, this led to 79 genus-level-congruent virus-host linkages and 293

**Data availability statement:** All sequencing read files were deposited in SRA (PRJNA1252777 and PRJNA1254298). The MAGs used in this study are available at https://doi.org/10.5281/zenodo.13984630. The viruses used in this study have been combined from two sources: (1) https://doi.org/10.5281/zenodo.8432681 and (2) DDBJ/ENA/GenBank under the accession number QGNH00000000. The scripts used in this study are available on https://doi.org/10.5281/zenodo.17459142. The accession numbers of the isolates and the phages are in S1 Table.

**Funding:** Funding was provided by the U.S. Department of Energy (https://www.energy.gov/; award #DE-SC0023307) to M.B.S. This material is based upon work supported by the National Science Foundation (https://www.nsf.gov/) under Grant Numbers #2022070 and #2149505 to M.B.S., and the National Institutes of Health (https://www.nih.gov/) grant R01AI169865 to M.B.S. The funders had no role in the study design, data collection and analysis, decision to publish, or preparation of the manuscript.

**Competing interests:** The authors have declared that no competing interests exist.

**Abbreviations:** AUC, area under the curve; CFUs, colony-forming units; MAGs, metagenome-assembled genomes; MOI, multiplicity of infection; OD, optical density; PFUs, plaque-forming units; PC, pseudoalteromonas-cellulophaga; ROC, receiver operating characteristics; SynComs, synthetic communities; vOTUs, viral operational taxonomic units.

new ones revealed by Hi-C alone, i.e., providing many new virus-host interactions to explore in already well-studied climate-critical soils. Overall, these findings provide empirical benchmarks and methodological guidelines to improve the accuracy and reliability of Hi-C for virus-host linkage studies in complex microbial communities.

## Introduction

Microbes play central roles in biogeochemical cycling and ecosystem functioning across Earth's systems [1]. Beyond their contributions to environmental processes, microbes support plant productivity [2], affect animal development [3], and influence human health, behavior, and disease susceptibility [4]. However, these microbial impacts are shaped by interactions within dynamic communities, including competition, symbiosis, and responses to predators. Towards the latter, viruses that infect microbes have emerged as prominent ecosystem components that modulate microbial activity and evolution. Viral infections can reprogram host metabolism into distinct 'virocells' with altered functional roles, induce lysis that regulates short-term population dynamics, and mediate horizontal gene flow that alters long-term evolutionary trajectories [5–8].

Despite the significance of virus-host interactions, most environmental viruses remain unlinked to their microbial hosts. Metagenomic sequencing has significantly expanded virus catalogs across diverse ecosystems, such as oceans, soil, and human-associated microbiomes [9–15], but host identification lags far behind. For instance, of the 15 million viruses cataloged in the IMG/VR database, only 7% have predicted host associations and merely 11% of those are experimentally validated [9]. This is because cultivation-based approaches, the gold standard for establishing 'infection', are time consuming and limited to cultivable microbes, which represent a tiny fraction of natural diversity [9,10]. Experimental alternatives such as molecular PCR [11,12] or probe-based techniques that co-detect virus and host markers inside cells [13] enable virus-host linkage, but are low throughput, target specific, and dependent on prior marker knowledge. Computational alternatives, such as tools like iPHoP [14] and VirMatcher [15], provide scalable solutions by leveraging genomic features like CRISPR spacer matches, k-mer profiles, and prophage signals. However, these bioinformatic approaches have two issues: (i) they struggle with novel viruses and (ii) experimental validation remains limited [10].

Proximity ligation sequencing via High-throughput Chromosome Conformation Capture (Hi-C) has recently emerged as a scalable and broadly applicable approach for experimentally inferring virus-host linkage. Hi-C for virus-host linkage inferences has now been applied in at least 10 studies spanning diverse ecosystems [16–25] and led researchers to conclusions that have never been observed in laboratory systems including that viruses can cross-infect phyla and even domains [20–22]. Conceptually, Hi-C uses proximity ligation to capture DNA molecules that are physically co-localized within intact cells by cross-linking them using formaldehyde. This process preserves spatial proximity between molecules (like viral genomes and host DNA) during active lytic virus infection or prophage integration. The cross-linked DNA

is then fragmentated with restriction enzymes, re-ligated to form chimeric DNA sequences, and subjected to high through-put sequencing (reviewed in [26]). Resulting reads can be used to identify virus-host linkages based on chimeric connections between viral and host genomic fragments. Two main analytic approaches are used: 1) read recruitment, where Hi-C reads are aligned to metagenome-assembled viral and host genomes and linkages are inferred based on the frequency and pattern of chimeric reads connecting viral and host contigs(e.g., [16,18,20,21]), and 2) binning-based strategies, which use Hi-C contact maps to cluster physically associated contigs, where co-binning of virus and host contigs suggests a potential relationship (e.g., [17,24]).

Despite the growing use of Hi-C for identifying virus-host interactions, several challenges remain in interpreting these linkages. *First*, the associations inferred from Hi-C data have not been experimentally validated. While one study demonstrated plasmid-host linkage accuracy using a synthetic community (SynCom) composed of *Lactobacillus brevis* and its two plasmids [27], comparable benchmarking for virus-host interactions is absent. *Second*, bioinformatic analyses of Hi-C data remain largely unstandardized and unoptimized. Of the 10 published virus-host Hi-C studies, only one conducted benchmarking and even then, it was limited to an *in silico* SynCom and relied solely on CRISPR spacer matches, without integrating additional computational features [24]. *Finally*, the broader lack of standardization in computational workflows (including differences in mapping strategies, thresholds, and filtering criteria) limits reproducibility and cross-study comparability. Together, these limitations highlight the need for systemic evaluation and methodological standardization to improve the reliability of Hi-C for virus-host linkage inference.

Here, we designed a SynCom composed of marine bacteria and their viruses with known virus-host interactions to systematically assess Hi-C performance for virus-host linkage inference. We used these SynCom experiments to develop analytic methods using the read recruitment approach to improve accuracy and reduce false-positive associations. Additionally, we applied the optimized Hi-C approaches to soil communities and compared the resulting virus-host predictions to those generated by a gold-standard computational host prediction tool. This dual application provides a foundation for standardizing Hi-C based virus-host linkage and offers a comparative dataset for improving prediction strategies in complex microbial communities.

## Materials and Methods

### Strains and growth conditions

Four bacterial strains were used (S1 Table): *Cellulophaga baltica* strain 18 was isolated from the Baltic Sea in 2000 (hereafter CBA 18; [28]), while *Pseudoalteromonas* sp. H71, H105 and 13–15 were isolated from the North Sea in 1990 (hereafter PSA H71, PSA H105, and PSA 13–15; [29]). Nine phages were used: five infecting *C. baltica* strains and four infecting *Pseudoalteromonas* strains (S1 Table). The CBA phages, φST, isolated on CBA 3 (strain not included in this study); φSM, isolated on CBA 3; φ46:1, isolated on CBA 46 (strain not included in this study); φ14:2, isolated on CBA 14 (strain not included in this study); and φ18:1, isolated on CBA 18, were isolated from the Baltic Sea between 2000 and 2005 [28]. The PSA phages, PSA-HM1, isolated on PSA H71; PSA-HS4 isolated on PSA H105; PSA-HP1, isolated on PSA H100 (strain not included in this study); and PSA-HS2, isolated on PSA 13–15, were isolated from the North Sea in 1990 [30].

Bacteria were grown on pseudoalteromonas-cellulophaga (PC) plates (20.5 g Sigma Sea Salts, 1.5 g peptone, 1.5 g proteose peptone, 0.6 g yeast extract, 13 g agar/L) at room temperature (RT). Single colonies were inoculated and grown stationary at RT overnight in PC liquid growth medium (20.5 g Sigma Sea Salts, 0.75 g peptone, 0.5 g yeast extract, 0.25 g proteose peptone, 0.25 g casamino acids, 1.5 mL glycerol/L). Phage strains were stored in phage buffer (20.5 g Sigma Sea Salts/L) and plaque-forming units (PFUs) were enumerated using the agar overlay method [31] with 3.5 mL molten soft agar (20.5 g Sigma Sea Salts and 6 g low melting point agarose/L) and 300 μl overnight bacterial culture per plate. Pairwise adsorption kinetics experiments were conducted to determine the adsorption performance of each phage to each strain (S2, S3 Tables).

## Adsorption kinetics experiments

Each phage–host pair was tested to determine which phages were able to adsorb to which host strains (S3 Table). Each bacterial strain was inoculated and transferred as for the synthetic community sampling experiments. Once each strain had reached mid-exponential phase and at least $1 \times 10^8$ CFU/mL, adsorption experiments were performed for each phage–host pair in duplicate. Cells and phage were incubated together in 1 mL PC medium at a multiplicity of infection (MOI) of 1 ($1 \times 10^8$ CFU/mL and $1 \times 10^8$ PFU/mL). A no bacteria control was also performed with each phage to ensure phage titer did not decrease due to other factors (e.g., sticking to the tube). For phages with no known host in the synthetic community (phages φ14:2 and φSM) or known poor adsorption to mock community bacteria (phage φST), their hosts of isolation were included as a positive control. Total and free phage were sampled and plated to determine PFU/mL at 0 min and 60 min, or 30 min where the known latent period is less than 60 min (phages HM1 and φ46:1). Adsorption was determined by comparing the free phage concentration at 60 min or 30 min to that at 0 min. A $t$ test was performed and any result that showed a significant decrease in free phage over time was considered positive for adsorption for that phage–host pair.

Follow-up adsorption kinetics experiments (S3 Table) were conducted on phage–host pairs showing significant adsorption in the initial experiment above. These experiments were performed in biological triplicates at MOI = 0.1 to reduce the chances of multiple adsorptions to a single host cell, and sampled every few minutes for ~30 min, but otherwise similar to the above. Some experiments did not include a 15 min time point – for these, data from 16 min is given instead.

## Synthetic community sample collection

Three synthetic communities (SynCom-1, SynCom-2, and SynCom-3) were made, each with three replicates. Accession numbers for all genomes used across SynComs are provided in S1 Table. All phage concentrations were based on infectious particles only, either when infecting bacterial hosts contained in the synthetic community or, for phages with no host in the synthetic community, on the host of isolation. Synthetic community 1 (SynCom-1) had equal numbers of each phage, whereas SynCom-2 and SynCom-3 had uneven numbers; final concentrations of each bacterial strain and phage are given in S1 Table. Overnight cultures of each bacterial strain were transferred into fresh PC media and grown stationary at RT until mid-exponential phase and a concentration of $>1 \times 10^8$ colony forming units (CFU)/mL was reached. Concentration was determined by reading the optical density of each culture at 600 nm and using a regression equation of OD versus CFU/mL. A total of $5 \times 10^8$ CFUs from each mid-exponential bacterial strain were combined in a 50 mL tube.

The nine phages were then added to each tube at the multiplicity of infection (MOI) indicated in S1 Table. After 15 min to allow phages to adsorb to their bacterial hosts, the community was diluted to 50 mL with PC phage buffer to slow adsorption and give a final cell concentration of $1 \times 10^7$ CFU/mL per strain. Aliquots of 1 mL were sampled, and flash frozen in liquid nitrogen. Additional aliquots of each replicate for SynCom-1 were preserved prior to flash freezing with 6.5% v/v DMSO, 6% w/v betaine, or 20% v/v glycerol. Samples were stored at −80°C until processing. Cell recovery rates after freeze-thaw cycle were determined through CFU count and qPCR quantification of free DNA after freeze-thaw cycle (S4 Fig).

## Cell recovery experiment after freeze-thaw cycle

Strains H71, H105, and 13−15 were inoculated in PZM, and strain 18 in MLB, and shaken overnight at 150 rpm at room temperature. Inoculations were transferred to fresh PC media and incubated stationary at room temperature. The optical density (OD) was monitored and graphed every 30−60 min until it reached $1 \times 10^8$ cells/mL. Each transfer was then diluted to $1 \times 10^7$ cells/mL in PC phage buffer. 1 mL of each dilution was pipetted into cryotubes with different preservatives (DMSO, betaine, glycerol, and no preservative, as in the SynCom sampling). Sets with preservatives were flash frozen in liquid nitrogen and stored at −80°C. Colony-forming units (CFUs) were plated from the no-preservative set, remaining samples were centrifuged, pellets were resuspended in PBS, filtered, and the resuspended pellets and filtrates were frozen at −20°C. Finally, CFUs were counted (S4 Fig).

## Free DNA test

We used qPCR to determine the concentration of free bacterial genomic DNA after pelleting of each strain before and after freeze-thaw in various preservatives (S4 Fig). Each strain was inoculated and transferred as for the synthetic community sampling experiments. Once each strain had reached mid-exponential phase and at least 1e8 CFU/mL, we performed the freeze-thaw experiment in various preservatives in triplicate. Each strain was diluted to 1e7 CFU/mL and preserved as follows: 1) no preservative, 2) 6.5% DMSO, 3) 0.5M betaine, or 4) 20% glycerol, as well as 5) a fresh (non-frozen), no preservative control. Sample sets 1–4 were flash frozen in liquid nitrogen and stored at −80 °C. Sample set 5 was centrifuged for 10 min at 10,000g at 4 °C to pellet cells and the supernatant discarded. The pellets were resuspended in 1 mL PBS, and 0.5 mL was 0.2 μm filtered to remove cells. Both the filtrate and remaining 0.5 mL of unfiltered pellet resuspension were stored at −20 °C. Sample sets 1–4 were thawed and treated as set 5. qPCR was performed on a 7,500 Fast Dx Real-Time PCR System (Applied Biosystems) with PerfeCTa SYBR Green FastMix Reaction Mix with low ROX (QuantaBio) in 10 μl reactions. Per reaction, we used 5 μl PerfeCTa master mix, 0.3 μl 10mM forward primer, 0.3 μl 10mM reverse primer, 3.4 μl nuclease-free water, and 1 μl template.

The target PSA H71 has a forward sequence of TTCTGATTCTGATGCGCGTG and a reverse sequence of CTTCTGAT-GGATTAGCGCCG, with an annealing temperature of 63°C. The target PSA H105 has a forward sequence of TGTATC-GCCTGCTTCACCTA and a reverse sequence of GCAGAACTTCCTACTTCCAGC, also with an annealing temperature of 63°C. The target PSA 13–15 has a forward sequence of GAGTTTGTGTCGTTGGATCGT and a reverse sequence of CCCAACTAGTAAACCACCAATCA, with an annealing temperature of 54°C. Lastly, the target CBA 18 has a forward sequence of TTTTACGAGAACGCCATCTTTCCAC and a reverse sequence of TGATGTAAGAGGGTTGAGGGCT, with an annealing temperature of 62°C.

Reactions were performed in technical duplicates with a standard curve consisting of 6x 10-fold dilution series of known concentration of the target strain DNA, which was used to calculate target sequence copies/μl. Cycling conditions were as follows: polymerase activation for 5 min at 95 °C; 40 cycles of 20 sec at 95 °C, 15 sec at primer annealing temperature (see SX), and 24 sec at 72 °C, and a 65 °–95 °C melt curve. Results were used to compare the number of DNA copies in the filtrate (free DNA) to that in the unfiltered, resuspended pellets (total DNA). The results from the frozen samples (sets 1–4) were compared to the fresh samples (set 5) to determine how free DNA changed with freeze-thaw for each preservative.

## Peat sample collection and preparation

In order to evaluate Hi-C performance in virus-host linkage from natural samples, we used three peat samples collected from a permafrost gradient in Stordalen Mire, Sweden. One sample was selected from each habitat: palsa, bog, and fen, all collected in 2014. The samples were processed as described previously [32]. Briefly, a soil core was collected from the surface (1–5 cm) for fen, bog, and palsa habitats, flash frozen in liquid nitrogen, and kept at −80°C until processing.

To establish a threshold based on known interactions, SynCom-1 was spiked into each peat sample at 1% of the total cell count. This concentration was chosen to minimize disruption of the native microbial community. The average cell count of the peat samples was $4 \times 10^8$ cells per gram soil, based on qPCR-based cell abundance data [33]. Before Hi-C library preparation, 0.5 g of each peat sample was suspended in 2.5 mL nuclease-free water and vortexed for 5 min, pelleted at 500g for 5 min. An aliquot of SynCom-1 (62 μL, approximately $2 \times 10^6$ cells) was added to the supernatant to achieve a 1% spike-in.

## Hi-C sample library generation

SynCom samples were thawed, and both these and the peat supernatants containing spike-ins were centrifuged to remove extracellular DNA. Resulting pellets were resuspended in the crosslinking solution provided in the ProxiMeta kit (PhaseGenomics). Hi-C libraries were prepped from the resuspended solution for each replicate of the synthetic

community or peat sample, following the manufacturer's instructions. Samples were pooled and sequencing was performed by SeqCenter (Pittsburgh, PA, USA) on an Illumina NovaSeq X 10B 300cyc lane, with the one exception of sample SynCom-1 replicate 1, which was sequenced at PhaseGenomics (Seattle, WA, USA) on an Illumina NovaSeq 6,000. Raw sequencing files (fastq) of SynComs and peat samples are available in the NCBI Sequence Read Archive (PRJNA1252777 and PRJNA1254298, respectively).

Unfortunately, for SynCom-1 preservative experiments, sequencing failed for one of the betaine-treated SynCom-1 replicates, and all glycerol-treated SynCom-1 samples. While successful samples yield over 100 million read pairs, these failed samples produced fewer than 1 million reads. The sequencing libraries from these samples had comparable yields to successfully sequenced libraries, and the cause of failure remains unclear. Further, for SynCom-3, sequencing for replicate 3 failed, with few reads (under 5 million). Unlike the preservative-treated samples above, these failed presumably due to low biomass in the prepared sequencing library. Because of the low sequencing output, these samples were excluded from further analysis.

## Hi-C data processing (synthetic communities)

The read recruitment approach was selected for this study because the binning approach produces finalized bins that would exclude the negative controls present in the SynCom. Raw reads from each sample were quality-filtered using BBduk (BBmap suite) [34] to remove the adapters and PhiX reads, and to trim and filter out low-quality reads. BWA-MEM [35], with the −5SP flag, was used to align Hi-C reads from each sample to the reference genomes of SynCom members. Since the number of aligned reads depends on the length of the contig, number of restriction sites and the sequence coverage, normalization of raw linkage counts was performed. MetaCC [36], a binomial regression-based normalization tool, was used with the enzyme flags Sau3AI and MluCI (the restriction enzymes included in the Proximeta kit) to perform this normalization. Subsequent analysis was conducted using the sparse matrix output 'Normalized_contact_matrix' from MetaCC. For genomes split across multiple contigs, contact scores were aggregated to produce a single normalized contact score for each virus-host pair. These normalized scores, along with associated $Z$-score (see below), are reported in S6 Table.

$Z$-scores were calculated for each SynCom replicate using the following equation:

$$Z = \frac{x - \mu}{\sigma}$$

where $x$ is the contact score of a specific virus-host pair, $\mu$ is the mean contact score of all virus-host linkages in the sample, and $\sigma$ is the standard deviation of those contact scores within the same sample.

Sensitivity and specificity of the predictions were calculated using the following equations:

$$Sensitivity = \frac{True\ Positives}{True\ Positives + False\ Negatives} \times 100$$

$$Specificity = \frac{True\ Negatives}{True\ Negatives + False\ Positives} \times 100$$

Host-phage interaction networks (S4 Fig) were constructed in R using the igraph package. Interaction data were split by sample, and directed graphs were generated from host-phage edge lists, with edge weights representing contact scores and edge colors indicating prediction accuracy. Nodes were classified as hosts or phages, and graphs were visualized using the Large Graph Layout ('layout_with_lgl'). To focus on high-confidence interactions, a filtered dataset retaining only edges with $Z$-scores > 0.5 was processed similarly.

 

For clarity, after reference mapping, we obtain read counts for each contig (associated with a MAG or virus). MetaCC then converts these read counts into *contact scores* by normalizing them based on the number of restriction sites, genome length, and sequencing coverage. A contact score represents the Hi-C interaction metric between a virus and a host. We refer to any virus–host pair with a contact score as a *linkage*. Throughout the manuscript, we distinguish between *raw linkages*, which include all virus–host pairs with a contact score, and *filtered linkages*, which are those retained after applying a *Z*-score-based filtering step.

**Hi-C data processing (peat samples)**

To process the Hi-C metagenomic samples from peat, we utilized existing databases of metagenome-assembled genomes (MAGs) [37] and viral operational taxonomic units (vOTUs) [38,39] previously generated from the Stordalen Mire ecosystem. While it would have been possible to assemble MAGs and vOTUs directly from the specific peat samples analyzed, we opted to use datasets derived from multiple samples collected across the same region. This approach was intended to enhance comprehensiveness and improve the likelihood of detecting host–virus linkages. For the MAG dataset, the MAGs were clustered at 95% average nucleotide identity, resulting in a total of 2,073 non-redundant MAGs. In order to combine the two viral datasets, we used CheckV v0.8.1 [40] to cluster the vOTUs from each dataset at 95% nucleotide identity and minimum 80% coverage, yielding 5,828 total vOTUs. These were combined with host and viral genomes from the SynCom spike-in community to generate a comprehensive reference database for read recruitment.

Read recruitment and downstream processing using MetaCC followed the same protocol as applied to the SynCom samples. Normalized linkage scores and *Z*-scores for each genome are provided in S9 Table. It was common for individual viruses to be associated with multiple predicted hosts. This is not unexpected, as many viruses are known to infect more than one host species. However, to reduce complexity in downstream comparative analyses (see below), we retained only the top host prediction for each virus. Specifically, we included all normalized scores greater than zero to calculate *Z*-scores across potential virus–host linkages. For each virus, the host with the highest *Z*-score was selected as the primary linkage. Note that for viruses that were detected in multiple peat samples, we included them as separate instances.

**Taxonomic comparisons**

To ensure consistency in taxonomic predictions across different methods, we standardized our approach wherever possible. The Stordalen MAG V2 dataset was classified using GTDB-Tk v2.3.2 [41] classify, with the r214 database and the option "--skip_ani_screen". For taxonomy determination via Hi-C linkage (i.e., vOTU to MAG connections), we applied the taxonomic annotations from the linked MAGs.

We used iPHoP v1.3.3 [14]) for virus-host prediction, conducting two separate: once with the default database ("Aug_2023_pub_rw"), and once with a custom database including the 95% ANI-clustered representative MAGs from the Stordalen Mire V2 dataset. For the default run, we executed the iphop predict command, using the default database without modification. For the custom database, we first processed the Stordalen Mire MAG representatives with GTDB-tk, using the de_novo_wf workflow and recommended parameters. The resulting output was formatted into an iPHoP-compatible database using iphop add_to_db. Finally, we then ran iphop predict with this custom database. iPHoP produces two outputs: one that integrates predictions from both phage-based and host-based tools, and another that includes only host-based predictions. To obtain higher taxonomic resolution, we used the output incorporating host-based tools, which provides direct connections to MAGs capable of species-level classification. In contrast, the phage-based and host-based tools output typically yield genus-level predictions only (see iPHoP Bitbucket documentation for details). We combined the host-based results from both the default and custom runs. Since individual viruses can have multiple high-confidence hits (defined as confidence scores ≥90), we selected the top hit for each virus. In cases where multiple top hits had equivalent scores, one was chosen at random.

VirMatcher (version 0.4.29; [15]) was run with default parameters. Based on prior work, a confidence score of ≥3 was used as the threshold for high-confidence predictions. Compared to iPHoP, VirMatcher produced considerably fewer predictions for unique viruses (510 versus 18, respectively). This discrepancy stems in part from the fact that many of the iPHoP linkages originated from public reference databases, while VirMatcher only utilized the Stordalen Mire MAGs. Due to the limited number of predictions generated by VirMatcher—particularly those overlapping with Hi-C linkages—we did not conduct further comparative analysis between VirMatcher and other tools. Overlap between iPHoP and Hi-C predictions is presented in S12 Table, and the overlap between VirMatcher and Hi-C predictions is in S13 Table.

For all analyses above, we used the top hit (or the first hit in cases of multiple top hits). To ensure robustness, we also tested: (1) all high-quality predictions, defined as those with Hi-C Z-scores ≥ 0.5 and iPHoP scores ≥ 90; (2) near-top predictions—those within 20% of the top Hi-C score or within ±2 points of the top iPHoP score; and (3) top predictions only (including ties). Results are shown in S8 Fig. VirMatcher was excluded from testing due to its substantially fewer predictions. When comparing methods at each taxonomic level, instances where both methods reported NA (no classification) were treated as NA, while cases where one method had NA and the other a value were considered disagreements.

For comparisons to Hwang and colleagues 2023, we used the published virus-host linkages from S10 Table from [20]. From that table, we took the "MAX_Linkage" values, converted them to Z-scores, and selected the highest Z-score for each vMAG present in the study. The calculated Z-scores are in S14 Table.

All scripts used for data preparation, analysis, and figure generation are available on Zenodo (https://doi.org/10.5281/zenodo.17459142).

## Results and discussion

### Evaluating Hi-C performance using a SynCom

To evaluate the accuracy of Hi-C in identifying virus-host interactions, we designed an experiment using a SynCom composed of four well-characterized bacterial strains and nine phages (which are viruses that infect bacteria) with known relationships and infection efficiencies [12–14] (Figs 1A, S1; S1, S2 Tables). In this SynCom, *Cellulophaga baltica* strain 18 (CBA 18) has three infecting phages, and three *Pseudoalteromonas* strains (PSA 13–15, PSA H105, PSA H71) that could be infected by 2, 1, and 1 of these phage strains, respectively. Moreover, two additional *Cellulophaga baltica* phages that do not infect any strains were added to the SynCom as 'distractor' phages. This SynCom mix of bacteria and phage

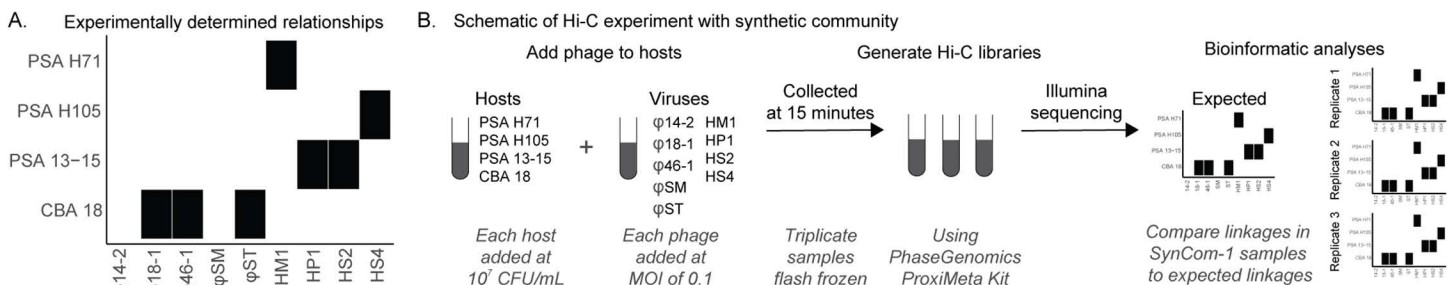

**Fig 1. Synthetic communities and experimental schema used to assess Hi-C virus-host linkages. A.** Synthetic communities were built from four bacterial strains (CBA, *Cellulophaga baltica*; PSA, *Pseudoalteromonas*) and 9 phages (listed fully in S1 Table) that were experimentally evaluated for infection in pairwise combinations via traditional plaque assays. Black boxes denote that the virus successfully plaques on the bacterial strain, whereas white (missing) boxes denote a negative, non-plaquing interaction. **B.** Schematic representation of the Hi-C experiment used to test virus-host relationships. After generating the synthetic community with the organisms mentioned above, Hi-C libraries were prepared and sequenced. Subsequently, bioinformatic analyses were performed to determine whether the expected virus-host linkages known from pairwise isolate-based experiments (denoted with black boxes) were observed.

strains was chosen to examine how varying infection dynamics influenced Hi-C performance, including infective and non-infective phages relative to the host community, and these quantitatively defined interactions were then used as the basis for Hi-C performance assessments.

To initially test Hi-C performance, we first constructed "SynCom-1", where bacteria and infective phage particles were added at high concentrations ($10^7$ CFU/mL for bacteria and $10^6$ PFU/mL for phages) (Fig 1B, S1 Table). The multiplicity of infection (MOI) of 0.1, higher than common natural levels, was chosen to optimize Hi-C performance while minimizing multiply-infected bacteria. Phage amounts were adjusted based on infection efficiency (determined by plating) to ensure equal quantity of infectious phages across strains (S1 Fig). After combining the phages with bacteria, we incubated the mixture for 15 min to allow adequate phage adsorption, as pairwise experiments showed 74%–99% adsorption within this time frame at MOI 0.1 (S3 Table). At the end of incubation, three technical replicates of SynCom-1 were immediately flash-frozen to halt phage activity (Fig 1B). Hi-C libraries were then prepared (PhaseGenomics ProxiMeta kit), and sequenced yielding 577 million 150-bp read pairs using standard Illumina sequencing approaches (see Materials and Methods). To minimize systemic biases, the aligned reads were normalized using MetaCC, a binomial regression-based tool that accounts for genome length, the number of restriction sites, and genome coverage [36], and produces normalized contact scores used for downstream analysis.

SynCom-1 analyses revealed both expected and unexpected phage-host linkages (prior to any data filtering). Expected linkages included all seven known interactions identified in prior single phage-host infection assays (Figs 2, S1 Fig), with 100% true positive detection (sensitivity) across three replicates (S6, S7 Tables). Unexpected linkages were for phage-bacteria pairs not known to infect that we included to represent negative controls (Fig 2, S1 Fig). For example, φ14:2 lacked a corresponding host in SynCom-1, and yet it showed linkages with multiple bacterial strains, and this finding persisted even across replicates (Fig 2B). Quantitatively, we found that the rate of true negative predictions (specificity) was low under these initial conditions, with an average across replicates of only 26% (S7 Table). Replicate reproducibility was high with replicates 1 and 2 sharing 30 identical interactions as compared to replicate 2 missing only one interaction and replicate 3 missing only six interactions (Fig 2B).

We next reasoned that the many false linkages might derive from the lack of a lower detection limit for the method, standard practice for new methods development for scientific disciplines at stage two (of four) or higher [42]. To this end, we evaluated contact score magnitudes as a proxy for virus-host linkage prediction confidence and evaluated where specificity declined. Ideally, we expected similar contact scores for the same linkages across replicates within SynCom-1. However, variability was observed for both correct and incorrect linkages. For example, CBA 18 and φ18−1 (a known interaction) had contact scores of 170, 277, and 13 (normalized per 100,000 reads to account for sequencing depth differences) in replicates 1, 2, and 3, respectively. Similarly, CBA 18 and HM1 (which do not have a known interaction), had contact scores of 41, 179, and 6 (normalized per 100,000 reads). Despite these variations, we observed that correct linkages generally had higher contact scores than incorrect ones (Fig 2A, black versus gray dots) such that 85% of the time, the highest contact score for each phage was to its correct host (S2 Fig). Nevertheless, contact score variability presents a challenge for establishing a universal threshold to minimize false positives solely from this metric.

Given this, we evaluated contact score performance in context with receiver operating characteristics (ROC) curves [42] to assess whether such models might improve the ability to distinguish between correct and incorrect linkages across replicates. Applying such approaches to SynCom-1 data resulted in an ROC curve with an area under the curve (AUC) of 0.71. This suggested that the contact scores can differentiate linkages (with AUC values closer to 1 representing better performance), but also that there was some room to improve upon the false positive rate (S3A Fig). While other studies had mitigated such noise via applying cut-offs for inter-genome and intra-genome ratios (e.g., a ratio of 0.6) or requiring a minimum number of linkages (e.g., 2 or 10 linkages), no statistical evaluations of effectiveness were conducted (see literature overview in S8 Table).

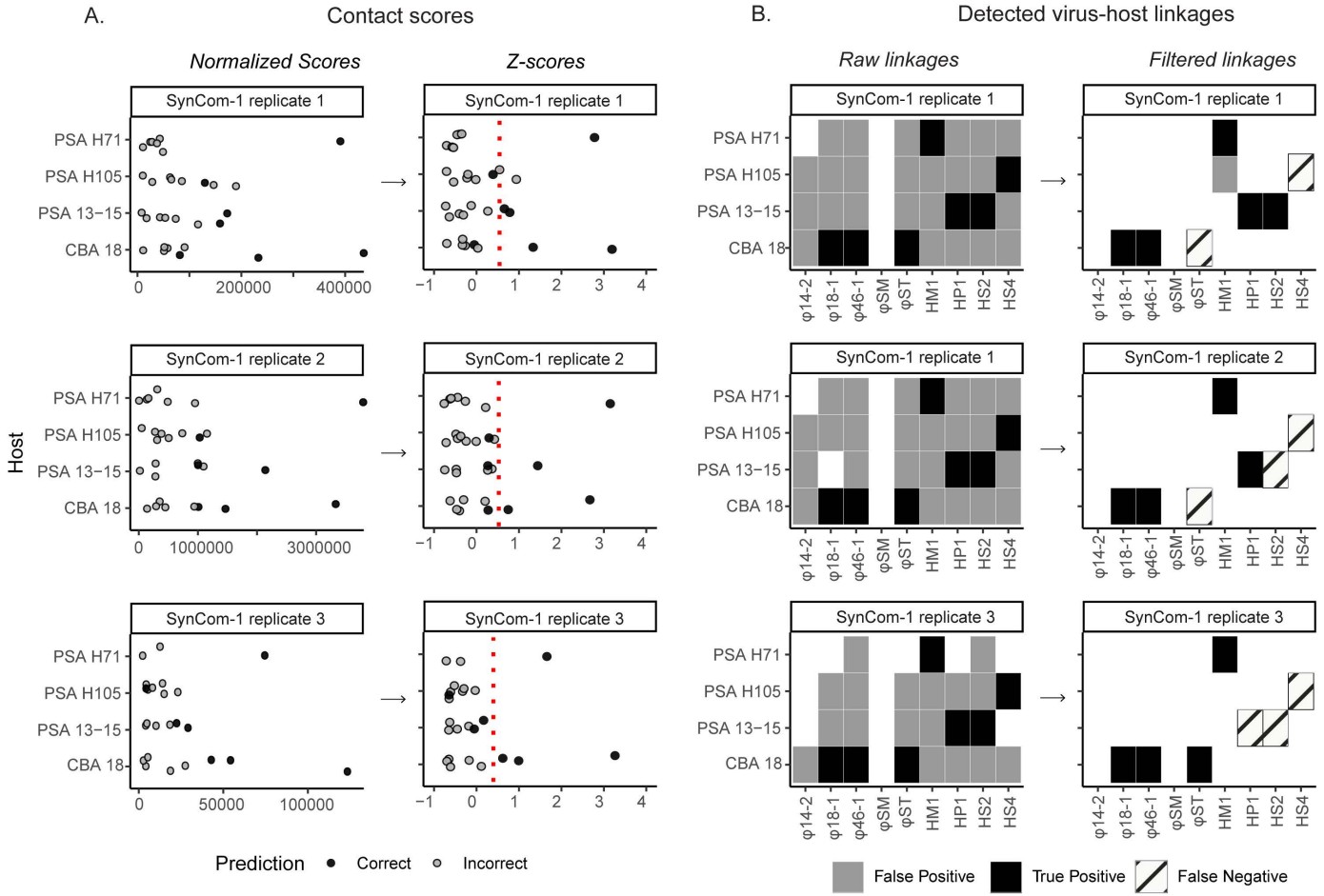

**Fig 2. Hi-C linkages from SynCom-1. A.** Contact scores (left) and corresponding Z-scores (right) calculated for each replicate of SynCom-1, categorized by host strains. The contact score represents the number of Hi-C linkages between a virus and a host genome, normalized for the number of restriction sites, genome length, and coverage. Z-scores were calculated from the contact scores within each sample to enable comparison across samples. The black dots indicate correct virus-host linkages, the gray dots indicate incorrect virus-host linkages. The red vertical dotted line is drawn at Z-score = 0.5. **B.** Virus-host linkages determined from a non-zero contact score (left) or using a filtering approach (i.e., requiring a Z-score generated from non-score normalized Hi-C scores above 0.5; right). The black boxes denote true positives, the gray boxes denote false positives, and the striped boxes denote false negatives. The data underlying this figure can be found in S6 Table.

To remedy this, we posited that Z-scores might be useful due to their value in RNA-Seq analyses for identifying differentially expressed genes [43] via considering each point relative to the mean. Transforming SynCom-1 contact scores to Z-scores revealed that most correct Z-scores were above 0 (Fig 2A), and this transformation improved the AUC to 0.93 (S3A Fig). The increased AUC (0.71 for contact scores to 0.93 for Z-scores) suggested that Z-scores more correctly identify virus-host linkages. Looking at the ROC curve further (S3B Fig), we selected a conservative threshold of 0.5 since it yielded the highest specificity across replicates. At this conservative threshold, 83% of virus-host linkages were removed with an impact of reducing average sensitivity across the replicates from 100% to 62%, while drastically increasing the average specificity from 26% to 99% (S7 Table). Moreover, this Z-score-based threshold showed consistent performance across all replicates, as further supported by network analyses (S4 Fig), which showed a substantial reduction in false positives while retaining true positives (pink and blue edges, respectively). While this is promising, we caution that further evaluation across future sample types and datasets will be valuable for assessing its universality, particularly for the specific Z-score cut-off used.

PLOS Biology

## Assessing Hi-C performance under different sample preservation conditions

With analytical optimization suggesting promise for Hi-C-based virus-host linkage approach, we recognized that vast archives of already-preserved samples might be an enticing opportunity to be repurposed towards this approach. Given this, we evaluated sample preservation conditions to help guide researchers on best practices. Also, anecdotal observations from our own work had suggested that nuances of preservation might be important as in the above experiments where we freeze-thawed the samples before Hi-C library preparation, we observed unexpected SynCom-1 linkages even after the analytical optimizations. For example, the correct linkages for PSA 13–15 (to HS2), PSA H105 (to HS4), and CBA 18 (to φST) had Z-scores under the threshold (Z scores < 0.5) (Fig 2). While the reduced linkage scores for φST can be attributed to its slow adsorption (i.e., not detectable within 15 min), the other phages exhibit faster adsorption (S2 Table) and so we expected to detect their linkages and yet did not.

Assuming possible freeze-thaw effects during sample preparation that impact linkage inferences, we hypothesized differentially affected recovery of intact versus leaky cells that could reduce true linkages and increase nonspecific linkages detected. To test this, we performed a freeze-thaw experiment to assess recovery of viable cells and DNA leakage. This revealed that PSA 13–15, PSA H105 and PSA H71 CFUs recovered poorly from freeze-thaw (1% on average, ranging from 0.1%–1.8%), whereas CBA 18 had approximately 100% recovery (S5A Fig). Additionally, free DNA recovered from post-freeze thaw samples was 14- to 17-fold higher than pre-freeze in unpreserved PSA strains, while that from CBA18 did not increase (S5B Fig). Together these findings suggest that PSA strains were more prone to leakage following freeze-thaw and suggest more broadly that host cellular integrity following freeze-thaw will variably impact Hi-C linkage performance in natural communities.

To assess whether cryoprotective agents could improve Hi-C performance by increasing cell viability and reducing leakiness, we evaluated commonly used cryoprotectants. Cryoprotective agents, such as dimethyl sulfoxide (DMSO) and glycerol, protect cells from cryoinjuries by inhibiting water crystallization, while betaine prevents osmotic rupture during the freeze-thaw cycle [43,44]. We first assessed whether treating with DMSO, glycerol, or betaine could improve cell viability and reduce leakiness prior to Hi-C library preparation and sequencing. Indeed, all three cryoprotective agents increased cell recovery (to 20% on average, 1%–60% range) for PSA 13–15, PSA H105 and PSA H71 (S5A Fig) and reduced cell leakiness (to 3.5-fold more free DNA than pre-freeze on average, 1.6 to 4.4-fold range) (S5B Fig). These results suggest that the cryoprotective agents improved cell viability in isolation, which may, in turn, enhance Hi-C performance in virus-host linkage.

Given this, we assessed SynCom-1 Hi-C performance using samples that were flash frozen compared to those preserved with DMSO, betaine, and glycerol (Fig 3). Only DMSO (3 replicates) and betaine (2 replicates) could successfully be Hi-C libraries prepared, while none of the 3 glycerol replicates were successful, possibly due to reagent incompatibility (see Materials and Methods). For those replicates that led to sequencing libraries and data, we examined the specific linkages between PSA 13−15 (to HS2), PSA H105 (to HS4), and CBA 18 (to φST) in the cryopreserved samples to determine if they improved. Despite having similar total linkages (S5 Table), there was no consistent improvement in the cryopreserved samples for these linkages. The only (slight) improvement was DMSO-treated SynCom-1 samples where PSA H105–HS4 linkages were detected above Z-score threshold in two out of three replicates. Together these data suggest that the tested cryoprotective agents do not drastically improve cryo-sensitive virus-host linkages, possibly due to their interference with Hi-C sequencing library preparation. While not evaluated directly here, we posit that chemical interactions between cryoprotective agents and Hi-C library preparation reagents may impact sample processing outputs.

Beyond specific linkages, we also examined whether there were major differences in overall linkages between the flash-frozen and cryoprotected samples to guide researchers on whether already-preserved archived samples might now be re-purposed for making virus-host predictions. Starting with raw linkages, we observed similar levels of non-self linkages across sample types (14% for betaine, 11% for DMSO, versus 11% for flash-frozen) (S5 Table). After filtering for Z-scores (Z ≥ 0.5), DMSO- and betaine-treated samples showed similar specificity to the flash frozen sample (96%, 97%,

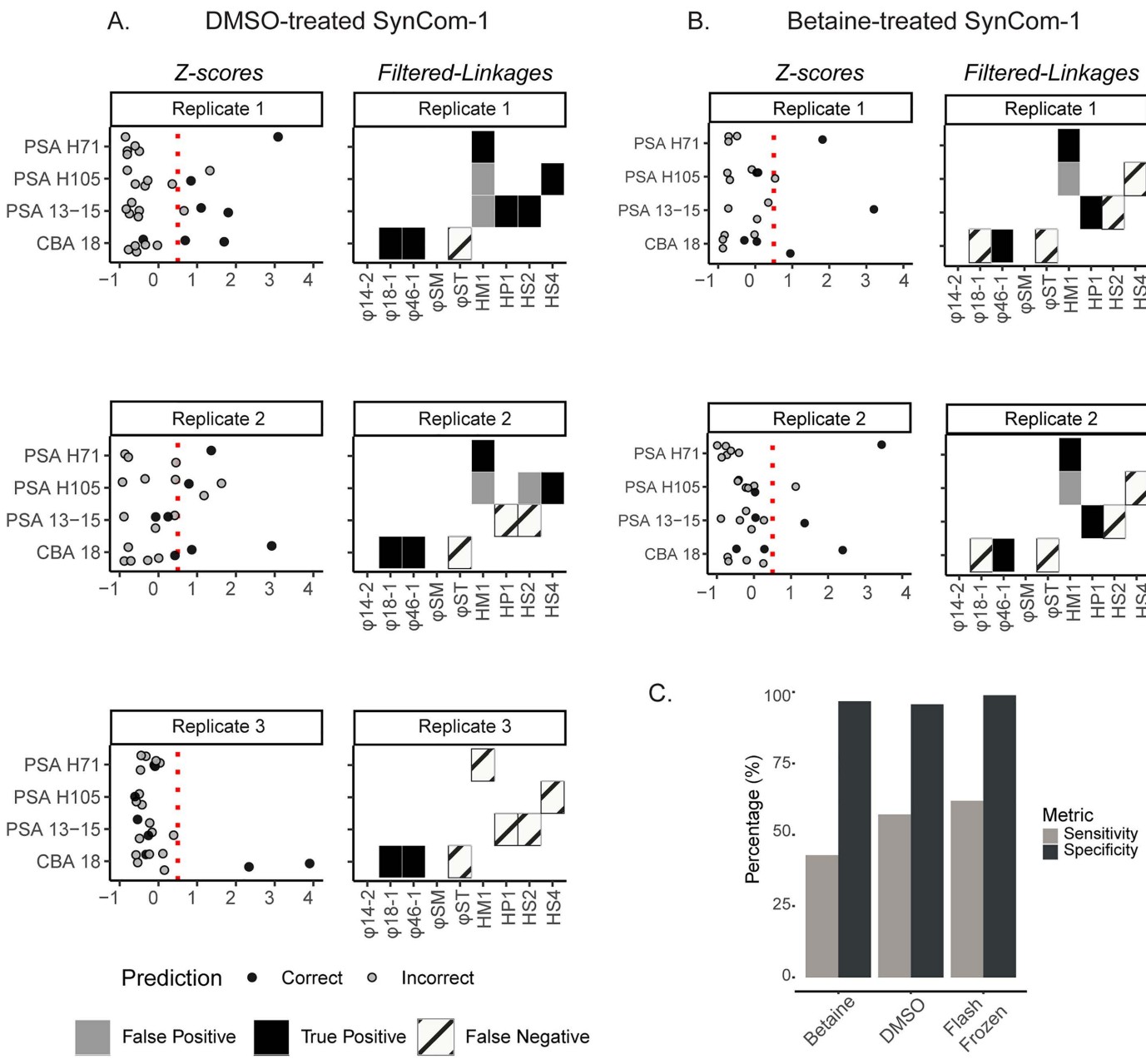

**Fig 3. Cryopreservation experiment to assess impact on SynCom-1 Hi-C linkages. A.** All Z-scores (left) and virus-host linkages (right) for each replicate of SynCom-1 cryopreserved with DMSO and categorized by host strains. Black and gray dots indicate correct or incorrect virus-host linkages, respectively, while black, gray, and striped boxes indicate true and false positives, and false negatives, respectively. The red vertical dotted line is drawn at Z-score = 0.5. **B.** Same data type as A, but for betaine-preserved samples. **C.** Average sensitivity (gray bar) and specificity (black bar) rates calculated for SynCom-1 treated with and without cryoprotective agents. The data underlying panel A and B can be found in S6 Table and the data underlying panel C can be found in S7 Table.

and 99% on average, respectively). However, sensitivity was generally lower in cryoprotective-treated SynCom-1 compared to flash-frozen samples. The betaine-treated samples showed clearer results (43% across all replicates), while the DMSO-treated samples had less consistent results (57%, ranging from 29% to 86%). In contrast, the flash-frozen samples

had more consistent results (62%, ranging from 57% to 71%) (Fig 3C). These data suggest that flash-frozen samples performed the best overall, as cryoprotectants might reduce sensitivity, but that DMSO- and betaine-treated samples do have value for later Hi-C analysis if those are the only sample options available.

## Evaluating the detection limit of Hi-C performance

Given the analytical and cryopreservation promise, we next sought to assess detection limits for Hi-C virus-host linkage inferences by examining flash-frozen SynComs with varying virus and host concentrations. This is because SynCom-1 phage concentrations were at $10^6$ PFU/mL with host-relevant MOIs of 0.1 (S1 Table), and such concentrations are relatively high for natural samples. To better reflect natural conditions, we created two new SynComs – SynCom-2 and SynCom-3 – where phage concentrations ranged from $10^3$–$10^6$ PFU/mL and MOIs ranged between 0.0001 to 0.1 as host concentrations were held constant at $10^7$ CFU/mL for each bacterial strain (S1 Table). Three technical replicates of SynCom-2 and SynCom-3 were attempted to assess reproducibility, though one replicate of SynCom-3 failed Hi-C proximity ligation sequencing post library preparation. Presumably, this was due to low biomass since this was the lowest concentration tested with DNA concentrations (0.070 ng/µL) near the lower threshold required for successful library generation according to the guidelines for the PhaseGenomics ProxiMeta kit.

From the five remaining successful Hi-C proximity ligation libraries, their resultant sequence data revealed clear detection limits. On the one hand, these libraries were comparable to those from SynCom-1 as an average of 14% and 13% of linkages were non-self in SynCom-2 and SynCom-3, respectively, which was only slightly higher than SynCom-1 (11%). On the other hand, problematically, $Z$-score threshold analyses ($Z \geq 0.5$) revealed that only one linkage passed the threshold in these latter SynComs across all the replicates (Fig 4). This one linkage was phage φ18:1 in SynCom-2 and phage φ46:1 in SynCom-3, and in both cases these phages were the only ones added at high concentrations of $10^6$ PFU/mL (MOI = 0.1). Other phages added at $10^5$ PFU/mL (MOI = 0.01) – φST in SynCom-2 and HS4 in SynCom-3 – had observable linkage data, but at $Z$-scores below the threshold ($Z < 0.5$) such that they were indistinguishable from incorrect linkages (AUC data in Supporting Information). Given that only the phages at $10^6$ PFU/mL had a detected linkage, there was an average of 100% specificity for both, but because of the absence of linkages for the other phages present in the SynComs, only 15% and 14% sensitivity was observed across SynCom-2 and SynCom-3, respectively.

Such a high minimum concentration required for accurate $Z$-score-based thresholding suggests that Hi-C-based virus-host linkages will be limited to the more abundant taxa or a higher MOI in natural samples and that researchers should consider $10^6$ PFU/mL a tentative lower detection limit. Additionally, although this study does not explore variations in host concentration, we hypothesize that lowering host abundance while maintaining a constant MOI would result in fewer overall infected cells and altered spatial dynamics between viruses and hosts. Together, these factors could reduce the number and diversity of virus-host linkages detectable by Hi-C. This rather high detection limit invites future methods optimization work if Hi-C-based virus-host linkage approaches are to have value across a diversity of samples in future work.

## Evaluating Hi-C linkages in natural communities

With analytical, cryopreservation, and detection limits now better understood, we sought to evaluate SynCom-derived $Z$-score thresholding approaches in natural communities. Specifically, we applied Hi-C analysis to flash-frozen permafrost thaw gradient samples collected from the well-studied, climate-critical model ecosystem, Stordalen Mire, in northern Sweden [33,44–46]. We selected three flash-frozen peat samples from surface layer soils of the dominant permafrost thaw stages at this site: palsa, bog, and fen (from least to most thawed). In response to recent recommendations for using spike-ins to enhance post-sequencing quantitative inferences [47–50], we then spiked these samples with ~$2 \times 10^6$ cells of SynCom-1 into each 0.5 g peat sample prior to Hi-C library generation. We estimated that this would represent approximately 1% of the total cells in these samples [32] (see Materials and Methods). Furthermore, since the SynComs originate from a different environmental system (marine), they are taxonomically distinct from the peat community and serve as

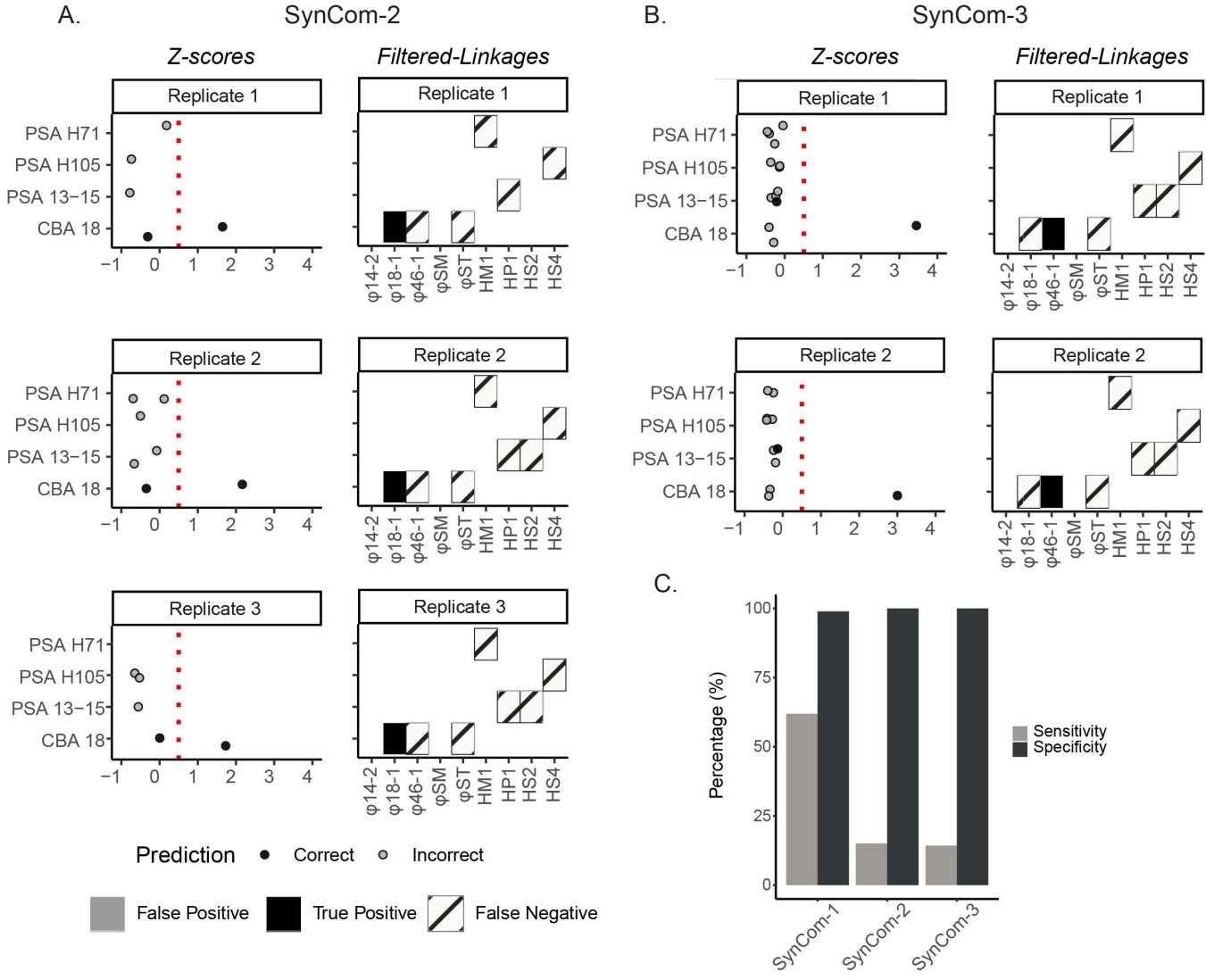

**Fig 4. Detection limit experiment to evaluate Hi-C linkages in varied concentration SynCom-2 and SynCom-3. A.** All *Z*-scores (left) and virus-host linkages (right) for each replicate of SynCom-2, categorized by host strains. All figure elements are the same as described in Fig 3. **B.** Same data type as A, but for SynCom-3. **C.** Average sensitivity and specificity rates calculated for SynCom-1, SynCom-2, and SynCom-3 without cryoprotective agents. The gray bar represents sensitivity, and the black bar represents specificity. The data underlying Fig 4A and 4B can be found in S6 Table and the data underlying Fig 4C can be found in S7 Table.

effective internal controls for thresholding. While environmental differences may introduce some bias, such as varying levels of complexity or relic DNA, the core Hi-C principle of capturing physical proximity within intact cells remains consistent across ecosystems.

We analyzed the resulting Hi-C peat sequencing datasets to identify virus-host linkages using the same method as applied to the SynComs. Briefly, we performed read recruitment to the spike-in genomes and a comprehensive dataset of de novo microbial and viral genomes including 2,074 metagenome-assembled genomes (MAGs) [37] and 5,828 virus-operational-taxonomic-units (vOTUs) generated from Stordalen Mire peat samples [38,39]. To address the microbial complexity inherent in soil samples, we performed deep sequencing to maximize linkage detection, applying 453M, 528M, and

326M total reads for palsa, bog, and fen samples, respectively. On average, 80% of the reads recruited to our reference genomes, with 0.001% [60,64] being chimeric virus-host sequencing reads deemed suitable for further evaluation. Unfortunately, despite the substantial sequencing, the spike-in experiment yielded few virus-host linkages from SynCom-1. Specifically, only two correct pairs (CBA 18 linked to φ18:1 and φ46:1) were detected in the bog sample, and one incorrect pair (a peat MAG to 14:2) in the fen sample, and all these pairs had low Z-scores (< 0.5). These findings demonstrate that while spike-ins are ideal for quantitative work [47–50], optimizing spike-in inocula for field samples, particularly in soils, presents numerous challenges such as the presence of humic substances that may inhibit DNA-based molecular reactions [51]. However, these data remained informative as follows.

First, comparison of normalized Hi-C linkage patterns across synthetic and natural peat communities revealed broadly similar proportions of virus-host associations, despite notable variation in linkage abundance and taxonomic predictions among habitats. Hi-C linkage patterns revealed similar proportions of virus-host linkages across peat samples and SynCom-1 (S6 Fig), indicating consistent linkage behavior between synthetic and natural communities. However, the number of virus-host linkages varied substantially by habitat with bog being the highest with 5,729 virus-host linkages (1 per 100,000 reads), followed by fen with 677 (0.2 per 100,000 reads), then palsa with 108 (0.02 per 100,000 reads) (S6 Fig). Virus-host pairs represented the smallest category of linkages, a pattern that was consistent across all peat habitats (S6 Fig). Such variation in linkages may reflect technical differences or underlying biological variation, such as differences in active viral infections (or prophage) activity across samples, known to vary in Stordalen Mire [39]. Of 6,064 virus-host linkages identified across peat samples, most vOTUs were linked to multiple contigs within or across MAGs. To reduce redundancy and spurious associations, only the highest normalized contact score per virus was retained, resulting in 495 unique virus-host associations spanning 19 microbial phyla (S9 Table). These linkages involved 30, 373, and 110 individual vOTUs from the palsa, bog, and fen habitats, respectively. Acidobacteria were the most frequently predicted host phylum (95 viruses), consistent with their prevalence in Stordalen Mire [38]. Host taxonomic profiles varied by habitat, likely reflecting habitat-specific differences in microbial community composition [38]. For example, while many phyla were shared among multiple habitats, Bacteroidota hosts were more predominant in the fen as evidenced by existing literature (S7A Fig). Although all 495 associations would be considered valid based on alignment-based Hi-C classification methods, further validation is needed to confirm their biological relevance.

Second, given the Hi-C-based virus-host linkages, we next aimed to assess how these associations compared to in silico host predictions to evaluate congruence. For in silico predictions, we used two tools that aggregate host predictions made from multiple approaches either via machine learning as in iPHoP [14] or a literature-derived importance valuation incorporated into a probabilistic model as in VirMatcher [15]. Our goal was to assess congruence at the species level where possible, to capture the most ecologically relevant taxonomic resolution. VirMatcher provides linkages from vOTUs to a user-submitted database (in this case, the Stordalen Mire MAGs), whereas for iPHoP we used host-based predictions only, which links vOTUs and a combined large public database alongside our custom addition of Stordalen Mire MAGs (see Materials and Methods). Of the 5,828 Stordalen Mire vOTUs analyzed, iPHoP and VirMatcher were able to make confident host predictions for 1,640 vOTUs (score ≥ 90) and 412 vOTUs (final score ≥ 3), respectively (S10, S11 Tables). Of these, 12% (200 vOTUs) for iPHoP and 4% (16 vOTUs) for VirMatcher also had a corresponding Hi-C linkage (Fig 5A, S12, S13 Tables). An additional 293 vOTUs were linked to hosts exclusively through Hi-C data (Fig 5A). However, when considering all normalized Hi-C contact scores, congruence between overlapping predictions was low. Only 43% (92 viruses) of iPHoP predictions and 67% (12 viruses) of VirMatcher predictions were congruent with Hi-C data at the genus level. Given that these predictions were supported across independent lines of evidence, we considered these congruent predictions to be higher-confidence virus-host linkages.

Third, to further understand congruence between methods, we focused on the iPHoP and Hi-C predictions, since there were more data available for analyses, and assessed congruence across taxonomic levels (Fig 5B). Without Z-score filtering, domain-level congruence was high (99%), but declined at finer taxonomic ranks, with phylum to family levels showing

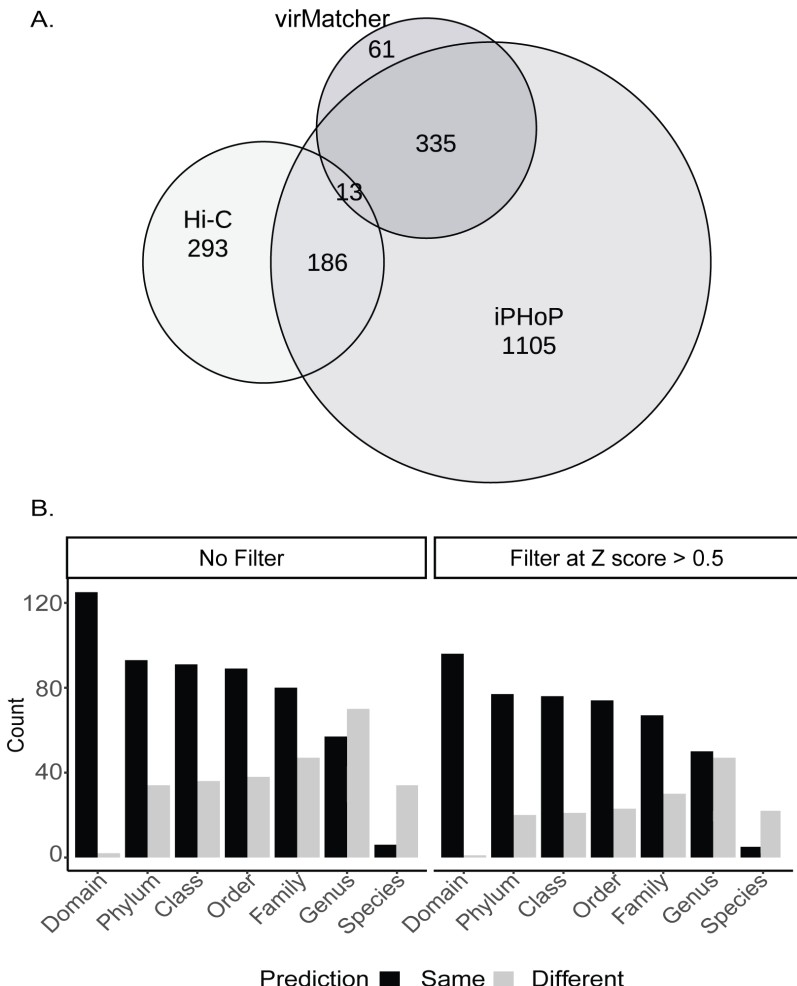

A. virMatcher
61
335
13
Hi-C
293
186
iPHoP
1105

B.

Prediction ■ Same ▪ Different

**Fig 5. Comparison of virus-host prediction from Hi-C and in silico tools. A.** Eular plot showing the overlap of viruses with host predictions obtained from the experimental Hi-C linkage approach, or one of two *in silico* tools (iPHoP and VirMatcher) that use different probabilistic models to aggregate output of various sequence-based features to create host prediction scores. **B.** Comparison of virus-host predictions across all samples between Hi-C and iPHoP, shown with and without applying a *Z*-score filter for the Hi-C linkages. Black bars indicate congruent predictions identified from both tools and gray bars indicate non-congruent predictions. Note: Although many viruses had multiple predicted hosts from each tool, only the top-scoring prediction for each virus was considered in this comparison. The data underlying Fig 5b can be found in S12 Table.

approximately 72% congruency and genus and species levels showing 43% and 15% (excluding the 155 predictions lacking a species assignment) congruence, respectively (S9 Fig). Applying a *Z*-score threshold (≥ 0.5) improved congruence, especially at lower ranks. Specifically, *Z*-score thresholding reduced the number of overlapping predictions from 214 to 142 and modestly improved congruency across most taxonomic levels – domain: 99%, phylum-class: 83%, genus: 48%, species: 18% (excluding the 104 predictions lacking a species assignment) (S10 Fig). Notably, higher *Z*-scores yielded more consistent host predictions (black bars, S10 Fig), especially at intermediate levels, while lower-confidence predictions (striped bars, S10 Fig) contributed more to mismatches. These results, which were robust to comparison strategies used (see Materials and Methods), suggest that while high-confidence computational predictions are largely consistent with Hi-C associations at broader taxonomic levels, they diverge at finer scales, particularly at the genus and species levels (S8 Fig). The latter is critical given such higher taxonomic ranks are where ecological relevance becomes more pronounced.

PLOS Biology

Discrepancies between Hi-C and bioinformatic predictions may stem from database biased training that iPHoP uses (e.g., favoring well-studied taxa) or Hi-C methodological artifacts (e.g., spurious contacts introduced during cross-linking or processing, differential membrane resistance to cross-linking chemistries, interference from relic DNA). While the latter are challenging to address experimentally (often requiring costly or currently limited single-cell approaches), we explored whether iPHoP's random forest model was trained on host genomes taxonomically representative of those found in our soil samples. Of the 48 phyla identified in our soil MAG dataset, only 7 were present in the iPHoP training set, yielding a weak correlation ($R^2 = 0.01$, p = 0.449; S7C Fig). This limited overlap suggests that training dataset database bias is at least one key factor underpinning the discordance between Hi-C and bioinformatic predictions. Nonetheless, when comparing virus–host linkages aggregated across phyla, iPHoP and Hi-C methods showed strong overall agreement ($R^2 = 0.852$, $p = 9.8 \times 10^{-10}$; S7B Fig), suggesting that while taxon-specific biases exist, they do not preclude high-level concordance between approaches.

Due to differences in experimental and bioinformatic methodologies across published studies that infer virus-host linkages using the Hi-C approach, direct comparisons of our Hi-C results to the literature are challenging. Robust cross-study comparisons would require reanalyzing all published datasets in a systematic manner, since almost all studies lacked a virus-host linkage table required for downstream analysis (S8 Table). However, one recent study [20] employed a similar approach as the one used in our study (using read recruitment to infer linkages and normalizing these linkages with binomial regression analysis) and provided the data in an analysis friendly format. Using these published data, we calculated Z-scores and applied our filtering based on highest linkages for each viral metagenome assembled genome (vMAG) (S14 Table). This revealed that only 30% (14 out of 47) of their reported vMAGs had predictions that exceeded the Z-score threshold ($Z \geq 0.5$). Notably, this removed cross-domain and cross-phyla interaction predictions, which contradict decades of experimental evidence on phage-host specificity. This finding underscores the need in the field for more conservative interpretation of virus-host linkage data in Hi-C based studies.

Assuming Hi-C proximity ligation becomes more broadly used, it becomes important to assess whether a Z-score threshold of ≥ 0.5 is broadly generalizable. While our SynCom data indicates that this threshold effectively removes spurious linkages, these results are based on controlled experimental conditions using marine phage-host experimental model systems. The foundational step in Hi-C proximity ligation is the use of a generalizable cross-linking molecular treatment to capture physically proximal DNA fragments, making the method theoretically applicable across diverse environments. However, natural systems may introduce variability that affects the reliability of this threshold. For example, soil environments may contain taxa with unique cell membrane structures, high concentrations of humic substances, and/or particle-associated relic DNA. All of these could complicate cross-linking efficacy or downstream molecular biology steps, whereas these issues might be drastically reduced in marine environments. Towards this, researchers working on the gut microbiome have already begun adapting key steps in the Phase Genomics ProxiMeta Hi-C kit protocol, such as increasing the volume of the crosslinking reaction and substituting restriction enzymes (e.g., using Dpn II instead of Sau3AI and MluCI) to optimize performance for their specific sample types [52].

To support the responsible application of Hi-C for exploring virus-host linkages, we offer the following recommendations for the field. At a minimum, Hi-C-based predictions should be compared against results from *in silico* tools, and any divergences, as demonstrated above, should be carefully examined. Wherever possible, we strongly recommend incorporating Z-score filtering guided by SynCom benchmarking. For applications in new environments or with under-characterized taxa, this means conducting SynCom-based benchmarking experiments, such as those done here. In cases where such benchmarking is not feasible, we recommend transparent discussion of the limitations of the Hi-C predictions and annotation of associated databases accordingly to avoid naïve propagation of dubious findings in future work. Finally, as with any measurement approach, there is a trade-off between specificity and sensitivity such that researchers should carefully consider the level of tolerance their research question requires. For studies making bold or novel claims, such as identifying cross-phyla or cross-domain linkages, a stricter false positive threshold is warranted. In contrast, for broader

community-level assessments, a more tolerant threshold might be acceptable. To assist with this in future studies, we have provided a centralized database (http://www.ivirus.us/proximity-ligation) for researchers to self-report a basic set of reporting requirements for the methods to have value.

## Conclusions

As viral ecology and the microbiome science continue to mature [42,53] so too must our standards to developing and applying new methodologies. This study seeks to optimize and benchmark Hi-C for establishing virus-host linkages, which, upon application, reveals that some previously published conclusions may be based on flawed or insufficiently validated applications. Hopefully, resetting the baseline for acceptable application of Hi-C virus-host linkage methods will promote more accurate and reproducible mapping of virus-host interactions. Such rigorously validated approaches are invaluable to inform practical applications, like phage therapeutics and microbiome engineering [54–56] in both environmental and medical contexts. Ultimately, improving our ability to identify virus-host interactions will enhance our capacity to leverage phages in our societal toolkit to address grand challenges like microbiome modulation for climate change mitigation [57].

## Supporting information

**S1 Fig. Efficiency of plating for members of the SynCom-1 community.** Here, the infectivity is derived from host range assays and is measured as plaque-forming unit (PFU) per mL. (PSA = *Pseudoalteromonas*, CBA = *Cellulophaga baltica*). The data underlying this figure can be found in S1 Data.
(TIF)

**S2 Fig. Contact scores for each phage in SynCom-1 replicate.** Black dots represent correct linkages and gray dots represent incorrect linkages. The data underlying this Figure can be found in S6 Table.
(TIF)

**S3 Fig. Evaluating sensitivity and specificity across Z-Score thresholds** . **A.** Receiver-operating characteristic (ROC) curve for normalized contact score (red) and *Z*-score (blue). **B.** Trade-off between sensitivity and specificity across *Z*-score thresholds for host-virus interaction predictions. The plot illustrates the relationship between sensitivity and specificity for six different *Z*-score thresholds used to filter host-virus interactions. Each line represents a distinct threshold: 0 (red), 0.1 (orange), 0.25 (green), 0.5 (light green), 0.75 (cyan), and 1 (blue). The x-axis shows specificity, and the y-axis shows sensitivity. The data underlying this figure can be found in S7 Table.
(TIF)

**S4 Fig. Network analysis of host-virus interactions in SynCom-1 across three technical replicates.** Each panel displays host-virus interaction networks for SynCom-1 replicates 1, 2, and 3. For each replicate, the left subpanel shows all detected linkages, while the right subpanel shows only statistically strong linkages with a *Z*-score ≥ 0.5. Nodes represent individual host or viral entities, labeled with the phage or host name. Edges between nodes indicate inferred interactions based on co-occurrence, with blue edges showing true positives and pink edges showing false positives.
(TIF)

**S5 Fig. Effects of preservation methods on bacterial viability and DNA integrity. A.** Viability of four bacterial strains (PSA H71, PSA H105, PSA 13–15, CBA 18) measured as colony-forming units per milliliter (CFU/mL) under different preservation conditions: Fresh Culture, Unpreserved, DMSO, Betaine, and Glycerol. **B.** Fold change in free DNA levels for the same strains under Unpreserved, DMSO, Betaine, and Glycerol conditions, indicating preservation-induced DNA release. The data underlying this Figure can be found in S2 Data.
(TIF)

**S6 Fig. Log-transformed read mapping pairs across contig and virus-host relationships in three peatland sample types.** Bar graph showing the number of read mapping pairs (log scale) for different contig pairs (dark blue), same contig pairs (medium blue), and virus-host pairs (light gray) across three sample types: Palsa, Bog, and Fen. This comparison highlights the variation in genomic linkage and potential virus-host interactions across distinct peatland environments. The data underlying this figure can be found in S3 Data.
(TIF)

**S7 Fig. Host predictions by taxonomy. A.** Host predictions (categorized at the phyla level) from Hi-C for peat samples. In each plot, the top bar shows viruses with any Hi-C linkage while the bottom bar includes only those with a Z-score ≥ 0.5. When multiple linkages were detected for a single virus, only the linkage with the highest normalized score was selected. Each phylum is represented by a different color, and the x-axis indicates the number of unique viruses. **B.** Taxonomic consistency between Hi-C and bioinformatic host-virus predictions. The scatter plot shows a linear regression comparing the number of phyla predicted by the bioinformatic method (iPHoP; x-axis) and the Hi-C method (y-axis). Each point represents a distinct phylum. The regression line demonstrates a strong positive correlation, with an $R^2$ value of 0.852 and a p-value of $9.82 \times 10^{-10}$, indicating that both methods yield similar taxonomic distributions. **C.** Comparison between phylum-level taxonomic counts in the MAG dataset from the soil community and the iPHoP training dataset. Each point represents a family observed in either dataset, with the x-axis showing counts from the iPHoP training data and the y-axis showing counts from Hi-C predictions from a natural community. The data underlying this Figure can be found in S12 Table.
(TIF)

**S8 Fig. Comparison of host predictions from iPHoP and Hi-C across three confidence thresholds, by habitat.** Bar plots show the agreement between host predictions from iPHoP and Hi-C across three habitats (palsa, bog, and fen). We used three different types of prediction filtering. Left column: All high-quality predictions (Hi-C Z-scores ≥ 0.5; iPHoP scores ≥ 90). Middle column: Near-top predictions (Hi-C Z-scores ≥ 0.5, all predictions with scores within 20% of the top score; iPHoP scores ≥ 90, all predictions with scores ±2 from top score). Right column: Top predictions (Hi-C Z-scores ≥ 0.5; iPHoP scores ≥ 90; only the top predictions, with multiple predictions if scores tied). The data underlying this figure can be found in S12 Table.
(TIF)

**S9 Fig. Comparison of host predictions (categorized at the phylum level) from Hi-C and iPHoP. The color represents whether the phylum-level host prediction was congruent or non-congruent between the two tools.** The left panel shows all phylum predictions that occurred at least five times in the dataset. The right panel shows the same, but only includes Hi-C predictions with Z score ≥ 0.5.
(TIF)

**S10 Fig. Breakdown of virus-host predictions from Hi-C and iPHoP, by congruency and Hi-C Z-score across taxonomic levels.** The figure summarizes the agreement between virus-host predictions from Hi-C and iPHoP across multiple taxonomic ranks (e.g., domain, phylum, class). The black bars indicate predictions that were congruent from both tools and the gray bars indicate non-congruent predictions. Solid bars and striped bars indicate predictions where the Hi-C linkage had a Z-score of 0.5 and above, and below 0.5, respectively. The data underlying this Figure can be found in S12 Table.
(TIF)

**S1 Table. Synthetic community composition and concentrations.** This table contains the full strain names (and abbreviated names) for each member of the synthetic community. Concentrations are listed in colony-forming units (CFU) per mL for hosts or plaque-forming units (PFU) per mL for phages. MOI was determined based on PFU/mL on the host of

isolation (not necessarily the respective host in the community). The * denotes phages that do not infect any members of the mock community. The # denotes the currently available genome version with an update to follow.
(XLSX)

**S2 Table. Phage–host adsorption results.** This table shows which phages adsorbed to which bacterial strains. Bacteria were grown to mid-exponential phase and incubated with phages at a multiplicity of infection (MOI) of 1. Bolded numbers show significant decrease in free phage over time (*t* test).
(XLSX)

**S3 Table. Adsorption kinetics experiment after 15 min.** The average percentage of starting phage adsorbed after 15 min is given, with standard deviation (SD), for each phage–host pair that showed significant adsorption based on the count of free phages present in the aliquot.
(XLSX)

**S4 Table. Information on sequenced samples.** Numbers shown for raw Illumina and quality-cleaned reads. The numbers reflect the number of paired reads.
(XLSX)

**S5 Table. Number of aligned pairs for each sample, categorized by alignment type.** "Total Reads Aligned" are the number of pairs that aligned to the reference genome, "Different Contig Pairs" are read pairs in which each read aligned to a different genome, "Same Contig Pairs" are the read pairs that aligned to only one genome, "Poor Match Pairs" are the read pairs that had a poor match quality in the alignment, and "Single Reads" are read pairs where only one read of the pair aligned to any genome.
(XLSX)

**S6 Table. Contact and Z-scores for SynComs Hi-C linkages.**
(XLSX)

**S7 Table. Sensitivity and specificity of the synthetic communities before and after applying a threshold and the area under the curve for SynCom-1, SynCom-2 and SynCom-3 replicates.**
(XLSX)

**S8 Table. List of publications that utilize Hi-C to identify virus-host interactions.** Table contains DOI link, data availability, details on virus-host linkages, and descriptions of normalization and filtering steps used in each publication. A single asterisk (*) indicates studies that did not clearly reference an accession number or a specific repository for the raw Hi-C reads, MAGs, or viruses. A double asterisk (**) denotes studies lacking a table that directly links viruses to hosts with a scoring system.
(XLSX)

**S9 Table. Contact and Z-scores for peat Hi-C linkages.**
(XLSX)

**S10 Table. iPHoP predictions for Stordalen Mire vOTUs.**
(XLSX)

**S11 Table. VirMatcher predictions for Stordalen Mire vOTUs.**
(XLSX)

**S12 Table. Comparison of Hi-C and iPHoP predictions.**
(XLSX)

**S13 Table. Comparison of Hi-C and VirMatcher predictions.**
(XLSX)

**S14 Table. Hydrothermal Vent Linkages with Z-Score.** The Z-score was calculated based on MAX_linkage for each rep_vMAG.
(XLSX)

**S1 Data. The numerical data underlying S2 Fig.**
(XLSX)

**S2 Data. The numerical data underlying S5 Fig.**
(XLSX)

**S3 Data. The numerical data underlying S6 Fig.**
(XLSX)

## Acknowledgments

We thank the EMERGE field team and Dr. Fen Li for the collection and the management of the soil samples. We thank Dr. Ben Bolduc, Dr. Garrett Smith and James Riddell for bioinformatic assistance, Dr. Yuxuan Du for MetaCC troubleshooting assistance and discussions, Dr. Olivier Zablocki and Dr. Suzanne Hodgkins for manuscript preparation assistance and data management assistance. We also thank Dr. Karna Gowda for his valuable advice on statistical inferences. Discussions at the Ohio State International Viromics Workshop (2024) were also helpful for working through literature nuances and anecdotes.

## Author contributions

**Conceptualization:** Rokaiya Nurani Shatadru, Natalie E. Solonenko, Christine L. Sun, Matthew B Sullivan.

**Formal analysis:** Rokaiya Nurani Shatadru, Natalie E. Solonenko, Christine L. Sun.

**Funding acquisition:** Matthew B Sullivan.

**Investigation:** Rokaiya Nurani Shatadru, Natalie E. Solonenko.

**Resources:** Matthew B Sullivan.

**Supervision:** Matthew B Sullivan.

**Visualization:** Rokaiya Nurani Shatadru.

**Writing – original draft:** Rokaiya Nurani Shatadru, Natalie E. Solonenko, Christine L. Sun, Matthew B Sullivan.

**Writing – review & editing:** Rokaiya Nurani Shatadru, Natalie E. Solonenko, Christine L. Sun, Matthew B Sullivan.

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
