## [Editor Report · Decision Letter 0]

30 Apr 2025

Dear Dr Sullivan,

Thank you for submitting your manuscript entitled "Synthetic community Hi-C benchmarking provides a baseline for virus-host inferences" for consideration as a Research Article by PLOS Biology.

Your manuscript has now been evaluated by the PLOS Biology editorial staff, as well as by an academic editor with relevant expertise, and I am writing to let you know that we would like to send your submission out for external peer review.

Once your full submission is complete, your paper will undergo a series of checks in preparation for peer review. After your manuscript has passed the checks it will be sent out for review. To provide the metadata for your submission, please Login to Editorial Manager (https://www.editorialmanager.com/pbiology) within two working days, i.e. by May 02 2025 11:59PM.

Kind regards,

Melissa

Melissa Vazquez Hernandez, Ph.D.

Associate Editor

PLOS Biology

---

## [Decision Letter · Decision Letter 1]

24 Jun 2025

Dear Dr Sullivan,

First of all, thank you for your patience while your manuscript "Synthetic community Hi-C benchmarking provides a baseline for virus-host inferences" was peer-reviewed at PLOS Biology. It has now been evaluated by the PLOS Biology editors, an Academic Editor with relevant expertise, and by three independent reviewers. I would like to apologize for the delay in sending you a decision.

In light of the reviews, which you will find at the end of this email, we would like to invite you to revise the work to address the reviewers' comments. Reviewer 1's recommendations refer to text or figure changes, among those are the inclusion of raw data examples and a network connection with host and virus. Reviewer 2 raises concerns regarding the novelty and significance of the results and would like some clarifications. Reviewer 3 wonders how the host concentration would impact the results, and if the comparisons between Hi-C-predicted and bioinformatically-predicted were inclined to specific taxonomic groups.

IMPORTANT: after discussion with the team, while the conceptual advance may be limited, we think that the study is an exceptionally rigorous exploration of the strenghts and limitations of a technique that can be a valuable resource for the community. As such, we would like to change the article type to a Meta-research type of article.

Given the extent of revision needed, we cannot make a decision about publication until we have seen the revised manuscript and your response to the reviewers' comments. Your revised manuscript is likely to be sent for further evaluation by all or a subset of the reviewers.

**IMPORTANT - SUBMITTING YOUR REVISION**

*Re-submission Checklist*

*Published Peer Review*

*PLOS Data Policy*

*Blot and Gel Data Policy*

Sincerely,

Melissa

Melissa Vazquez Hernandez, Ph.D.

Associate Editor

PLOS Biology

REVIEWERS' COMMENTS:

Reviewer #1: The manuscript systematically evaluates the performance of Hi-C technology in virus-host interaction studies, providing critical insights into its accuracy, detection limits, and sample preservation conditions through experiments involving synthetic communities (SynComs) and natural soil samples. The research design is rigorous, and the data analysis is comprehensive, offering valuable methodological references for the field of microbial ecology. However, certain aspects of the paper require further clarification and improvement to enhance its scientific rigor and readability.

Major comments:

1. The introduction paragraph is somewhat lengthy and scattered in focus. It is suggested to further refine the introduction to highlight the challenges of current virus-host association identification, the potential and shortcomings of the Hi-C method, and the innovation points of this study.

2. The figures need to be improved. The figures have low resolution and lack visual appeal. Such as ImageGP 2 (https://doi.org/10.1002/imt2.239 ) can generate high quality figures and with reproducible scripts.

3. While the paper notes the low consistency between Hi-C and bioinformatics prediction tools (e.g., at the genus and species levels), it does not fully explore the potential reasons for this discrepancy (e.g., database bias or methodological limitations). Further analysis is recommended. Regarding the generalizability of the Z-score threshold, the paper mentions the need for future validation but could more specifically discuss its potential applicability in other ecosystems or sample types.

4. Descriptions for some figures (e.g., Figures S5-S8) are relatively brief. It is recommended to explain their key information in more detail in the main text. The Venn diagram in Figure 5 could include specific numerical values or percentages to more intuitively illustrate the differences in consistency between methods. A better Venn diagram webserver EVenn is recommended https://www.bic.ac.cn/EVenn/ to show better and high-quality results.

5. If possible, it is recommended to include examples of raw data (e.g., alignment results of partial Hi-C reads) in the supplementary materials to enhance methodological transparency.

6. A network connection with host and virus is also recommended, such as iNAP 2 (10.1002/imt2.235) or ggClusterNet 2 are good tools for network analysis.

Minor comments:

1. Which kinds of Hi-C you used. Recently published GutHi-C in iMeta. What the similar or different with GutHi-C, please compare and explain in the discussion paragraph.

2. The terms such as "bacteriophage" and "phage" should be unified in the text to avoid confusion.

3. Some sentences are overly long and structurally complex (e.g., the first paragraph of the Introduction). Splitting or simplifying them is recommended to improve readability.

4. The paper mentions failures in Hi-C library preparation for certain samples (e.g., glycerol-treated samples) but does not explore potential reasons in depth. It is recommended to supplement related analyses or hypotheses, such as whether this is due to reagent compatibility or sample processing steps. For the weaker Hi-C signals observed in natural soil samples, potential biological or technical factors (e.g., soil complexity or DNA extraction efficiency) should be discussed.

5. Labels and annotations in Figures 2 and 3 could be more explicit, such as directly marking "true positives" and "false positives" with specific examples.

6. It is recommended to use a ROC curve as a supplementary illustration to the Z-score distribution plot to enhance the intuitiveness of the evaluation criteria.

The manuscript makes a significant methodological contribution, providing practical technical guidance and benchmark data for virus-host interaction research. With the aforementioned improvements, its scientific value and reader-friendliness can be further enhanced. It is recommended for a major revisions.

Reviewer #2:

Although the study question is quite actual and the study design and methodology are overall valid, the significance and novelty of the results presented do not appear to be corresponding to the level of the journal. In other regards, below are some specific comments:

Abstract:

"Microbiomes influence diverse ecosystems, but viruses increasingly appear to impose

key constraints." - why "but" rather than "and" ?

Typo: "virushost"

"optimized protocols" - Worthy specifying if the optimization refers to the wet-lab part or rather the data analysis alone.

"Hi-C inferred virus-linkages with in silico bioinformatic predictions" - Isn't the former a subset of the latter? Maybe the latter was supposed to be more specific (i.e. sequence-based predictions).

Intro

Line 44: "including humans where they can influence cravings" - It is not clear why the term "cravings" is listed #1 among the roles of human gut microbiomes and what that means.

55: missing punctuation "oceans (9-11) lakes".

90: "Hi-C method remains largely unoptimized" - Please specify if you are referring to the wet-lab part or bioinformatics or both here.

101: Please specify what unites the bacterial taxa selected as SynComs' components in terms of their environmental niche. Are they all marine? This is worthy being mentioned in the Abstract as well.

Also please explain why the niches were different for the SynComs and the real-life communities - marine vs. soil. Could it have served as a bias - i.e. the principles that worked for marine microbes being possible different from the soil ones?

Materials and Methods

232: "we retained only the top host prediction for each virus" - Were the cases when multiple hosts scored (almost) equally high in comparison with the rest treated specially? If not, then is not the approach unstable in defining the host in such situations, affecting the results and conclusions?

Results

310: What was the cutoff threshold value for separating the signal from noise?

Provided three of the SynCom bacterial genomes represent the same species, their genomes might be very similar, and particular genomic fragments might be identical between the three. Did the authors check if such fragments exist? If yes, they could have contributed to the arise of false-positives.

The same question applies to the SynCom phages.

In the view of the comment above and overall based on the principles of the methodology, it appears appropriate to not even count the low-level contacts as linkages.

Regarding the "unexpected linkages". Were the phage-bacteria in this experiment checked via an alternative approach (e.g. qPCR, microscopy)? If not, could there be a possibility that phages manifested an extended host range in the specific artificial community and conditions?

Fig. 2A: It might make sense to reduce the extent of the point jitter along the Y axis, as some points start approaching the points of another taxon.

Fig. 2B: Based on the description of the fills, the figure leaves an impression that the "Raw linkages" are way inferior in terms of the number of FP linkages. However, no threshold had been applied, unlike the case of Z-score. Would not it be a fair visual comparison if an optimal threshold were selected for "raw linkages" (based on the analysis of AUC, F1 score, G-means or other metric) and the respective plots were binarized akin to "Filtered linkages"?

326: Typo: "HI-c"

354: "most correct and incorrect linkages had Z-scores above or below 0, respectively" - Not sure if this phrase is informative, provided linkages belong to either correct or incorrect classes, no surprise their scores are non-zero.

357: "Looking at the ROC curve further (S3 Fig), we selected a conversative threshold of 0.5 since it yielded the highest

specificity and sensitivity across replicates." -- It is not clear how the value 0.5 was obtained from the shown curves. Overall, while visual inspection is valid for initial assessment, the analytical definition requires strictness: which criterion was used to select this specific threshold as optimal? (e.g. the one that optimizes F1 score, or G-means, or other metric). It's not possible to get the highest specificity and sensitivity simultaneously.

Fig. S3: How were the data from 3 replicates summarized to get a single AUC? Please show the variability across the replicates, as well as the variability of the optimal threshold.

454: For the SynCom-2 and -3, please show that the optimal threshold previously obtained using a single community (SynCom-1) is still optimal (or at least performs well) - e.g. via an AUC analysis. Could it be the case that it's not optimal anymore suggesting lack of generalization and a need to compute an optimal threshold in a alternative way for each new sample?

Reviewer #3 (Katherine McMahon):

Review of PBIOLOGY-D-25-01349_R1

Summary

This manuscript is an elegant effort to benchmark the approach of using the Hi-C method to infer virus-host pairs in multi-species assemblages. The authors used pure cultures of bacteria combined into synthetic communities, along with a collection of phage, to determine how well the Hi-C method recapitulates the experimentally validated ability of each phage to enter (infect?) each host. They examined the influence of multiplicity of infection on Hi-C sensitivity and specificity, evaluated several steps in the experimental protocol (both wet-lab and data-processing/normalization), and compared how Hi-C based host-virus predictions performed relative to bioinformatics approaches. Finally, they applied what they learned to environmental samples. Overall, the authors determine that the Hi-C approach is promising but should be used in combination with bioinformatics approaches, and with attention to detection limits and internal controls (e.g. spiked synthetic communities).

The manuscript is well-written, clear and accessible to read, and comprehensive in its description of the experimental design and findings. I'm sure the authors had hoped for more definitive results (and better performance of the Hi-C), but it is a solid contribution with insight that other researchers hoping to use this method in environmental samples, will greatly appreciate.

I have only a few suggestions to improve the manuscript.

Major comments

The experiments conducted with varying MOI held the host abundance constant at 10^7 CFU/mL. As I understand it, this is 100x higher than typical seawater and 10x higher than freshwater (and hard to compare to soil/sediment/feces of course). I would like to have the authors add some speculation about how the HOST concentration would impact the results (beyond DNA yield), while holding the MOI constant. I'm not asking for more experiments, just a few sentences identifying this as a concern.

Towards the end of the manuscript I found myself wondering whether the comparisons between Hi-C-predicted and bioinformatically-predicted was skewed in any way towards particular taxonomic groups. This would be an easy analysis to do, and would satisfy other readers with similar curiosity.

Minor comments (intended to increase clarity and ease of interpretation):

Line 164: Pet peeve: this number has far too many significant figures. Use 4 x 10^7 instead.

Line 285-286. In my first read-through, the term "positive and negative interactions" was confusing (mostly the latter). I'm assuming you mean a host-phage pair that CAN interact (positive) and a pair that CANNOT interact (negative). The terms do make sense in the context of "false negative", so I don't know why it's confusing (but it is)! Consider re-wording to clarify, or at least explain?

Line 300: I think you are referring to Figure 1B.

Line 314: The term "negative interactions" confused me (see comment about line 285-286 above).

Lines 342-351: I suggest moving this paragraph to the supplementary material. I realize that it fits in the narrative of "first we tried this, then we tried that", but it doesn't add much to the main text since it's not mentioned in a main figure.

Line 443: The host concentration should be in units of CFU/mL, right?

Line 471: How does the detection limit nominally depend on host cell abundance? It seems like the MOI would be the more important metric, but maybe I am mis-interpreting.

Line 507: I think there are two words missing: "..varied substantially by habitat, WITH bog BEING the highest…"

Figure 1: Panel B, for the hosts, I think this should be CFU/mL not PFU/mL? I appreciated this figure, since it helped a lot to understand the setup.

Figure 2A: What is the meaning behind the normalized linkage score? Is it useful to compare across replicates, or is it only appropriate to compare within a replicate? I assume the latter, since that would motivate using the z-score. For people not familiar with Hi-C, it would help to explain this in the legend, tho. Also, what is the difference between a contact score and a linkage score? I suspect they are synonyms, so stick with one please.

---

## [Decision Letter · Decision Letter 2]

22 Oct 2025

Dear Matt,

Thank you for your patience while we considered your revised manuscript "Synthetic community Hi-C benchmarking provides a baseline for virus-host inferences" for publication as a Meta-Research Article at PLOS Biology. This revised version of your manuscript has been evaluated by the PLOS Biology editors, the Academic Editor and one of the original reviewers.

Based on the reviews and on our Academic Editor's assessment of your revision, we are likely to accept this manuscript for publication, provided you satisfactorily address the remaining editorial requests. Please also make sure to address the following data and other policy-related requests.

1) We routinely suggest changes to titles to ensure maximum accessibility for a broad, non-specialist readership, and to ensure they reflect the contents of the paper. In this case, we would suggest a minor edit to the title, as follows. Please ensure you change both the manuscript file and the online submission system, as they need to match for final acceptance:

"Benchmarking with synthetic communities provides a baseline for virus-host inferences from Hi-C proximity linking"

2) We do not have a word limit. Please move the Supporting Information to the main text which can provide the readers an easier access to all information.

Please supply the numerical values either in the a supplementary file or as a permanent DOI’d deposition for the following figures:

Figure 2AB, 3ABC, 4ABC, 5C, S1, S2, S3AB, S5AB, S6, S7ABC, S8, S10

4) Please cite the location of the data clearly in all relevant main and supplementary Figure legends, e.g. “The data underlying this Figure can be found in S1 Data” or “The data underlying this Figure can be found in https://doi.org/10.5281/zenodo.XXXXX”

5) Supplementary files (e.g., excel). Please ensure that all data files are uploaded as 'Supporting Information' and are invariably referred to (in the manuscript, figure legends, and the Description field when uploading your files) using the following format verbatim: S1 Data, S2 Data, etc. Multiple panels of a single or even several figures can be included as multiple sheets in one excel file that is saved using exactly the following convention: S1_Data.xlsx (using an underscore)

6) Please ensure that your Data Statement in the submission system accurately describes where your data can be found and is in final format, as it will be published as written there

7) Thank you for providing the underlying code in GitHub. However, because Github depositions can be readily changed or deleted, please make a permanent DOI’d copy (e.g. in Zenodo) and provide this URL in the manuscript and Data Availability Statement.

We expect to receive your revised manuscript within two weeks.

*Published Peer Review History*

*Press*

Sincerely,

Melissa

Melissa Vazquez Hernandez, Ph.D.

Associate Editor

PLOS Biology

REVIEWERS' COMMENTS

Reviewer #1:

The author's response has been fully addressed my concerns. The quality of the paper has apparently improved. I agree with the publication of this article.

---

## [Editor Report · Decision Letter 3]

5 Nov 2025

Dear Matt,

Thank you for the submission of your revised Meta-Research Article "Benchmarking with synthetic communities provides a baseline for virus-host inferences from Hi-C proximity linking" for publication in PLOS Biology. On behalf of my colleagues and the Academic Editor, Jeremy Barr, I am pleased to say that we can in principle accept your manuscript for publication, provided you address any remaining formatting and reporting issues. These will be detailed in an email you should receive within 2-3 business days from our colleagues in the journal operations team; no action is required from you until then. Please note that we will not be able to formally accept your manuscript and schedule it for publication until you have completed any requested changes.

PRESS

Sincerely, 

Melissa

Melissa Vazquez Hernandez, Ph.D., Ph.D.

Associate Editor

PLOS Biology
